# Midlife and old-age cardiovascular risk factors, educational attainment, and cognition at 90-years – population-based study with 48-years of follow-up

Anni Varjonen[1]*, Toni Saari[1], Sari Aaltonen[1], Teemu Palviainen[1], Mia Urjansson[1], Paula Iso-Markku[1,2], Jaakko Kaprio[1], Eero Vuoksimaa[1]

1 Institute for Molecular Medicine Finland (FIMM), HiLIFE, University of Helsinki, Finland, 2 HUS Diagnostic Center, Clinical Physiology and Nuclear Medicine, University of Helsinki and Helsinki University Hospital, Helsinki, Finland

* anni.varjonen@helsinki.fi

## Abstract

We examined the associations of midlife and old-age cardiovascular risk factors, education, and midlife dementia risk scores with cognition at 90+years, using data from a population-based study with 48 years of follow-up. Participants were 96 individuals aged 90–97 from the older Finnish Twin Cohort study. Individual cardiovascular risk factors assessed via questionnaires in 1975, 1981, 1990, and 2021–2023 included blood pressure, body mass index, physical activity, and cholesterol, and self-reported educational attainment. The Cardiovascular Risk Factors, Aging, and Dementia (CAIDE) score and an educational-occupational attainment score were used as midlife dementia risk scores. Cognitive assessments included semantic fluency, immediate and delayed recall from a 10-word list learning task, and a composite cognitive score. Regression analyses were conducted with dementia risk factors predicting cognition at 90+years, adjusting for age, sex, education, follow-up time, and apolipoprotein E genotype (ε4-carrier vs non-carriers). Results showed that higher education and higher educational-occupational score were associated with better cognitive performance in all cognitive measures. Those with high midlife blood pressure scored significantly higher in all cognitive tests than those with normal blood pressure. Conversely, those with high old-age blood pressure scored lower in semantic fluency and composite cognitive score, but not in immediate or delayed recall. Other cardiovascular risk factors and the CAIDE score did not show consistent associations with cognition. Education appears to have a long-lasting protective effect in cognitive aging, whereas midlife and old-age cardiovascular risk factors were not significantly associated with cognition at 90+years.

**Data availability statement:** Due to the consent given by the participants and easier identification of twin data, data is not publicly available. Data are available through the Institute for Molecular Medicine Finland (FIMM) Data Access Committee (DAC) for those with authorization, who have IRB/ethics approval and an institutionally approved study plan. Please contact the FIMM DAC (fimm-dac@helsinki.fi) for more details.

**Funding:** This work was supported by the Finnish Brain Foundation [to A.V]; Orion Research Foundation [to P.I.M]; The Biomedicum Helsinki Foundation [to P.I.M]; Juho Vainio Foundation [to S.A]; Academy of Finland Center of Excellence in Complex Disease Genetics [grant number 352792 to J.K]; NONAGINTA data collection was funded by the Research Council of Finland [grant numbers 320109, 345988 to E.V]; the Research Council of Finland [grant number 314639 to E.V], and the Sigrid Jusélius Foundation Senior Fellowship [to E.V]. The funders played no role in the design, execution, analysis, or interpretation of data, or writing of the study.

**Competing interests:** The authors have declared that no competing interests exist.

## Background

Those who are 90-years or older are the fastest growing population segment in many countries [1], with a high prevalence of cognitive impairment and dementia, estimated nearly at 40% [2]. Potentially modifiable lifestyle factors are associated with dementia risk differently across the lifespan with seemingly decreasing adverse effects towards older age [3–6]. There is strong evidence for education to serve as a protective factor against dementia [7]. Cardiovascular (CV) risk factors measured in midlife have been associated with increased dementia risk [7,8], but there is evidence that CV risk factors at old age are protective against dementia; for example, higher body mass index (BMI) and developing hypertension at old age have been associated with better old age cognition [3,6,9]. These paradoxical findings likely reflect reverse causation as the risk factors are often measured with short follow-ups during the preclinical stage of the dementia process, which starts even 20 years before the clinical diagnosis [10]. There is a need for studies where CV risk factors have been measured in middle age before the disease process has started, preferably with over 20 years of follow-up to rule out reverse causation [9].

Our aim was to use a population-based sample of twins with up to 48 years of follow-up to investigate if CV risk factors measured in midlife (age range 42–51) and very old age (age range 90–98), including BMI, blood pressure (BP), cholesterol, and physical activity (PA) predict cognition at nonagenarian age. We also examined education, and two previously validated midlife dementia risk scores, the Cardiovascular Risk Factors, Aging and Dementia (CAIDE) [11], and the educational-occupational score [12] in association with cognition at 90 years old. While these scores were originally developed and validated to predict late-life dementia based on midlife risk factors, our aim here was to explore whether these midlife profiles remain predictive of cognitive outcomes at very old age. We hypothesized that those with more midlife CV risk factors, and lower education, have poorer cognition than those with less midlife CV risk factors and higher education. We also expected that the CV risk factors measured at 90 years are not associated or have a reversed association with cognition on the basis that in later stages of dementia, certain risk factors such as low BMI and hypertension may no longer indicate risk, but rather reflect the progression of the disease itself [13].

## Materials & methods

### Participants

The participants were twins from a population-based older Finnish Twin Cohort (FTC) (established in 1974) [14] and participated in a sub study focusing on twins turning 90-years-old, referred to as the NONAGINTA –study. All eligible surviving twins were recruited for the study, regardless of whether their co-twin was deceased or did not participate. The data collection began June 1, 2020, and is still ongoing. For NONAGINTA enrolment, those who were 90 years old or older from the FTC (born before May 1930) were invited to participate, and invitations were sent when people had reached 90 years of age (those born in 1930–33). Data collection included

telephone interviews for cognition and postal questionnaires for lifestyle factors. We also used earlier FTC data that were collected through postal questionnaires with a baseline in 1975 (all twins), and follow-ups in 1981 (all twins) and 1990 (only those born 1930 or later were invited) (Fig 1). For our analysis, we included everyone who had participated by January 2024. For the NONAGINTA study, 187 people had participated in the postal questionnaire in 2020–2023 (27%), and of them 96 also participated in the telephone interviews (13 full twin pairs; 8 MZ, DZ). Of them, 93 participants also had baseline questionnaire data from 1975.

## Cardiovascular risk factors

The CV risk factors included BMI, BP, cholesterol, and PA. We also included education. These were self-reported measures assessed with a postal questionnaire at midlife in 1975 (age range 42–51) (baseline), 1981 (age range 48–58), 1990 (age range 57–60) and 2020–2023 (age range 90–98). Detailed descriptions of the CV risk factors and education are given in the Supplement (S1 Text).

BMI was calculated for 1975, 1981, 1990, and 2020–2023 based on self-reported weight and height (kg/m2) at the time of the questionnaire. We used mean BMI in 1975 and 1981, BMI in 1990, and at 90 years old in the analyses. In case of missing data from either 1975 or 1981 questionnaire, we used single value from the other questionnaire.

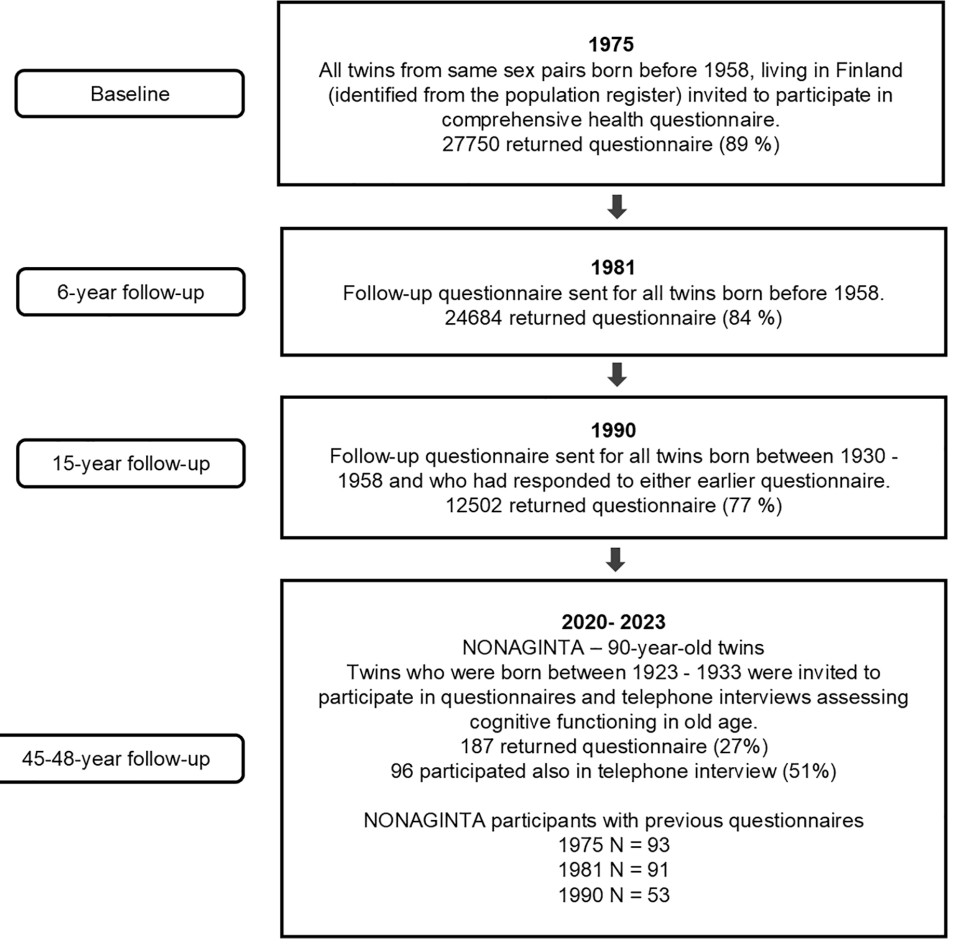

**Fig 1. Flow chart of the study.**

We used a binary BP variable, with high and normal BP groups, assessed in 1975, 1981, 1990 and 2020–2023. This was based on a self-report measure from the questionnaires (see the Supplement, S1 Text). Finnish guidelines for hypertension thresholds in 1970-1980s were set at 160/95–175/100 mmHg [15,16]. In 1975, 3 of the 6 participants and in 1981, 5 of the 13 participants reporting high BP used anti-hypertensive medications, respectively. Of those with normal BP, only one person out of 82 had used anti-hypertensives briefly in 1975, none in 1981.

A history of total cholesterol level was assessed in 1981, 1990, and 2020–2023 questionnaires (not included in 1975). Total cholesterol level was also transformed into a binary variable, with normal/low and high cholesterol groups.

PA was assessed as MET (metabolic equivalent of task) hours expended per day (continuous variable) [17]. We used the mean of 1975 and 1981 MET hours/day, MET hours/day measured in 1990, and MET hours/day measured at 90 years old in the analyses. In case of missing data from either 1975 or 1981 questionnaire, we used single value from the other questionnaire.

For descriptive purposes we used self-reported measures of the presence of doctor-diagnosed diabetes (yes/no) [8], frequency of alcohol use (S1 Table) and smoking status (yes/no) [18], in 1975, 1981, 1990, and at 90 years old.

## Education

Education was assessed with a self-reported measure in the questionnaires in 1975, 1981, and 2020–2023. For our analysis, we used a three-category education variable (6 years or less, 7–11 years, and 12 years or more). In the analyses, in each time point (1975, 1981, 1990, and 2020–2023) we used the education variable that was reported in the same time-point as the risk factors were measured. For 1990, education from 1981 was used. Missing values were filled in based on most recent questionnaire.

## CAIDE

CAIDE score was based on scores calculated in [8]. The score was determined from participants' self-reports in 1975, 1981, and 1990 questionnaires, including BP, BMI (from height and weight), cholesterol levels, and exercise frequency. When data were missing at one time point, values from an alternate year or later questionnaire (e.g., 1990 for cholesterol) were used. The classification thresholds and scoring followed those of the original CAIDE model, with minor adjustments to accommodate available data. Original CAIDE was developed using in-person measurements [11]. The CAIDE score ranges from 0–15, with higher values indicating higher dementia risk. See detailed description in the Supplement (S1 Text). More detailed explanation on individual risk factors transformed into a total CAIDE score can be found in [8].

## Educational-occupational score

The educational-occupational score was based on the scores calculated in [12]. It consisted of the following self-reported variables from the 1975 (mean age 45) and 1981 (mean age 52) postal questionnaires: age, years of education, work status (not working, homemaker, or working/ studying), complexity of work (very monotonous, somewhat monotonous/ variable, or very variable), physical loading of work (heavy manual labor, manual labor with lifting, light manual labor, and nonmanual work) and work environment (mainly outdoors or both indoors and outdoors, vs. indoors only) [8]. The questions for these variables and their categorization are explained in detail in [8].

## Cognitive measures

Three cognitive measures were used from a telephone interview-based cognitive assessment at age 90 years. Semantic fluency was measured with one-minute animal naming. Episodic memory was assessed with two measures including immediate and delayed recall of 10-word list from the modified Telephone Interview for Cognitive Status [19,20] using modification including three learning trials [21]. Immediate recall measure was total numbers of words recalled in trials

1–3. In delayed recall, participants were asked to repeat the words again after about 3 min from the immediate recall. Each of the three test scores were also standardized into z-scores, and their average was calculated to create a composite cognitive score, representing overall cognitive performance. The number of animals named, words recalled in trials 1–3 and after delay, and the composite cognitive score were used as dependent variables for semantic fluency, immediate, and delayed recall, respectively.

## Statistical analyses

We used linear regression analyses with semantic fluency, immediate recall, and composite cognitive score as dependent variables and individual CV risk factors, education, total CAIDE score, CAIDE without education, and educational-occupational score as independent variables. For the delayed recall, we used negative binomial regression due to distribution skewness towards zero values. Only complete cases were included in the analyses. We ran individual-level analyses (between-family) and adjusted for the non-independent data structure (family data). Regression analyses were conducted for each CV risk factor separately with three sets of covariates. Model 1 included age, sex, and follow-up time as covariates, model 2 included education as an additional covariate to model 1 covariates, and model 3 included *APOE* genotype as additional covariate to model 2 covariates. For *APOE* status, participants were categorized into ε4-carriers and non-carriers (more detail in the Supplement, S1 Text).

We conducted drop-out analyses (design-based F-test and two-sample t-test accounting for family relatedness) comparing midlife and very old age risk factors and dementia risk scores between different participation groups: those who completed both a telephone interview and postal questionnaire at age 90, those who completed only postal questionnaire, and those who were invited but did not participate. We also conducted an inverse probability weighting (IPW) analysis to adjust for potential bias due to non-random dropout. Logistic regression was used to estimate the probability of remaining in the study, with predictors including the educational-occupational score (which differed significantly between participants and non-participants), sex, and family relatedness (accounted for via clustering). Stabilized inverse probability weights were then applied in the regression models. All analyses were run on Stata MP 18 (64-bit).

## Ethical considerations

The NONAGINTA study was approved by the Coordinating Ethics Committee of the Hospital District of Helsinki and Uusimaa (HUS). The participants gave written informed consent for their participation. Returning questionnaires in 1975, 1981, and 1990 were considered as consent to participate and consistent with Finnish legislation on medical research at that time.

## Results

### Descriptive statistics

The descriptive characteristics are described in Table 1. The average follow-up time from baseline to NONAGINTA study was 45.92 years (range between 45–48 years). Mean age was 45 years (standard deviation (SD)=2.27) at the time of the 1975 questionnaire, and 91 (SD = 1.93) for questionnaire sent in 2020–2023.

The average number of words in semantic fluency was 15.20 (SD = 4.92), 11.34 (SD = 4.45) for immediate recall, and 2.21 (SD = 2.27) for delayed recall. The delayed recall measure showed a floor effect, with 29% of participants scoring zero. The mean composite cognitive score was −0.03 (SD = 0.75). One participant was excluded from the analyses for the composite score as an outlier, with a score more than three SD from the sample mean. Including the outlier in a sensitivity analysis did not alter the results. The participants had 8.97 (SD = 4.45) years of education on average. A total of 19 individuals were *APOE* ε4-carriers (22%).

For BP, 6% (6/91) reported having high BP in 1975, 15% (13/85) in 1981, and 22% (11/49) in 1990. In the 2020–2023 questionnaire, 70% (63/ 90) reported having high BP. For cholesterol, 19% (7/36) reported having high cholesterol in midlife (1981),

**Table 1. Descriptive statistics for those who participated in telephone interviews and questionnaires in 90-year-old data collection.**

| Characteristics All (90 years old) | N | All (96) | Men (41) | Women (55) |
|---|---|---|---|---|
| | | Mean (SD) | Mean (SD) | Mean (SD) |
| **Age** (years) | | | | |
| 1975 | 92 | 45.42 (2.27) | 45.84 (2.33) | 45.13 (2.21) |
| 1981 | 90 | 51.80 (2.43) | 52.08 (2.26) | 51.59 (2.23) |
| 1990 | 53 | 59.27 (1.11) | 59.31 (1.11) | 59.25 (1.12) |
| 2020-2023 | 96 | 91.22 (1.93) | 91.27 (1.95) | 91.20 (1.93) |
| **BMI** (kg/m$^2$) | | | | |
| 1975/1981 | 94 | 24.67 (3.05) | 24.77 (2.46) | 24.60 (3.44) |
| 1990 | 51 | 25.78 (3.72) | 25.43 (3.09) | 25.97 (4.06) |
| 2020-2023 | 95 | 25.15 (4.10) | 24.56 (3.50) | 25.60 (4.48) |
| **Blood pressure** (high/normal) | | | | |
| 1975 | 91 | 6/85 | 2/36 | 4/49 |
| 1981 | 85 | 13/72 | 4/34 | 9/38 |
| 1990 | 49 | 11/38 | 3/14 | 8/24 |
| 2020-2023 | 90 | 63/27 | 22/17 | 41/10 |
| **Cholesterol** (high/normal) | | | | |
| 1981 | 36 | 7/29 | 5/18 | 2/11 |
| 1990 | 42 | 23/19 | 6/10 | 17/9 |
| 2020-2023 | 65 | 31/34 | 14/17 | 17/17 |
| **Physical activity** (MET hours/day) | | | | |
| 1975/1981 | 88 | 2.40 (1.71) | 2.74 (1.85) | 2.14 (1.57) |
| 1990 | 52 | 3.24 (4.56) | 4.88 (7.15) | 2.36 (1.87) |
| 2020-2023 | 84 | 1.70 (1.65) | 1.79 (1.48) | 1.64 (1.79) |
| **CAIDE total** | 54 | 7.61 (1.92) | 7.57 (1.73) | 7.65 (2.13) |
| **EDU-OCU** | 94 | 16.77 (3.15) | 16.68 (3.28) | 16.83 (3.08) |
| **EDU 1975** (≤6yrs/ 7–11 yrs./ ≥ 12 yrs.) | 82 | 33/40/19 | 11/19/8 | 22/21/11 |
| **EDU 1981** (≤6yrs/ 7–11 yrs./ ≥ 12 yrs.) | 94 | 33/41/20 | 13/18/9 | 20/23/11 |
| **EDU 90 yrs.** (≤6yrs/ 7–11 yrs./ ≥ 12 yrs.) | 96 | 38/38/20 | 17/15/9 | 21/23/11 |
| **Semantic fluency** | 96 | 15.20 (4.92) | 15.73 (4.88) | 14.80 (4.95) |
| **Immediate recall** | 96 | 11.34 (4.45) | 11.02 (4.23) | 11.58 (4.63) |
| **Delayed recall** | 96 | 2.21 (2.27) | 1.88 (1.76) | 2.45 (2.57) |
| **Composite cognitive score** | 95 | −0.03 (0.75) | −0.04 (0.72) | −0.02 (0.78) |
| **APOE** status (ε4 carrier/non-carrier) | 83 | 20/65 | 5/31 | 15/34 |

APOE = apolipoprotein E, BMI = body mass index, CAIDE = Cardiovascular Risk Factors, Aging and Dementia score, EDU = education, EDU-OCU = educational-occupational score, MET = metabolic equivalent hours per day, SD = standard deviation, yrs. = years.

55% (23/42) in late midlife (1990) and 48% (31/65) in very old age (2020–2023). For midlife PA, mean MET hours/day were 2.40 (SD = 1.71), for late midlife 3.24 (SD = 4.56), and in very old age, the mean was 1.70 (SD = 1.65). The mean midlife BMI was 24.67 (SD = 3.05), late midlife was on average 25.78 (SD = 3.72), and in very old age the mean was 25.15 (SD = 4.10). The mean total CAIDE score was 7.61 (SD = 1.92) and the mean educational-occupational score was 16.77 (SD = 3.15).

In 1975 there were 11 current smokers, but only three in 1990 and as 90-year-olds. Nobody reported diabetes in 1975, 1981 or 1990 but among the 90 + year olds, 12 had type 2 diabetes. Most participants reported very little or no use of alcohol (S1 Table).

## Individual protective and risk factors

**Education.** All results for risk and protective factors and dementia risk scores are displayed in Fig 2 (semantic fluency), Fig 3 (immediate recall), Fig 4 (delayed recall), and Fig 5 (composite cognitive score). For semantic fluency, those with 12 or more years of education scored higher than those with 6 years or less (b = 4.48, 95%CI: 1.50; 7.44, p = 0.004) (reference group) and the result was similar when controlling for *APOE* (S2 Table). Those with 7–11 years of education (b = 2.63, 95%CI: 0.58; 4.68, p = 0.013) and those with 12 or more years of education (b = 5.36, 95%CI: 3.75; 6.98, p < 0.001) remembered more words in immediate recall compared to those with 6 years or less education. The results were similar when including *APOE* as a covariate (S2 Table). For delayed recall, those with 7–11 years of education and those with 12 or more years of education scored higher compared to those with 6 years or less education: b = 0.49, 95%CI: −0.01; 0.99, p = 0.06, and b = 1.11, 95%CI: 0.75; 1.46, p < 0.001. The result remained essentially the same for 12 or more years of education when including *APOE* as covariate (b = 1.04, 95%CI: 0.63; 1.45, p < 0.001) (S2 Table). For composite cognitive score, those with 12 years or more education had higher scores than those with 6 years or less (b = 1.09, 95%CI: 0.83; 1.34, p < 0.001), also when controlling for APOE (S2 Table). The results for composite cognitive score showed medium to large effect sizes: participants with 7–11 years of education had 0.41 SD higher composite cognitive score than those with 6 years or less, and those with 12 years or more education had 1.45 SD higher composite cognitive score than those with 6 years or less education.

## CV risk factors

Higher midlife BMI was associated with better semantic fluency (b = 0.29, 95%CI: 0.002; 0.58, p = 0.05), also with *APOE* as covariate (S3 Table). Midlife BMI was not significantly associated with immediate recall, delayed recall, or composite cognitive score (S3 Table). BMI in late midlife was not associated with any of the cognitive measures (S4 Table). Very old age BMI was not associated with any cognitive measures (S2 Table).

Midlife PA was not significantly related to any of the cognitive measures (S3 Table). A higher PA in late midlife was associated with better semantic fluency (b = 0.22, 95%CI: 0.04; 0.34, p = 0.02), also when accounting for *APOE*, but not with immediate or delayed recall, or the composite cognitive score (S4 Table). Very old age PA was also not associated with semantic fluency or immediate recall (S2 Table). Higher very old age PA was associated with lower scores in delayed recall (b = −.15, 95%CI: −0.28; −0.02, p = 0.02), also with *APOE* in the model (S2 Table). Higher PA in very old age was associated with lower composite cognitive score (b = −0.09, 95%CI: −0.17; 0.02, p = 0.019), but not when including *APOE* in the model (S2 Table).

Those who reported high BP in 1975 scored higher in semantic fluency (b = 3.10, 95%CI: 0.43; 5.77, p = 0.02), immediate recall (b = 3.27, 95%CI:0.33; 6.21, p = 0.03) and delayed recall (b = 0.87, 95%CI: 0.41; 1.32, p < 0.001) compared to those with normal BP, also with *APOE* in the model (S5 Table). Those with high BP also had higher composite cognitive scores (b = 0.81, 95%CI: 0.32; 1.29, p = 0.001), also with *APOE* as a covariate (S5 Table). The results for composite cognitive score showed a large effect size whereby participants with high BP had 1.08 SD higher composite cognitive score than those with normal BP. People with high BP in 1981 showed better semantic fluency: (b = 2.94, 95%CI: 0.18; 5.70, p = 0.04), but not with *APOE* as a covariate. There was no significant association for 1981 BP in immediate recall, but people with high BP scored higher on delayed recall (b = 0.68, 95%CI: 0.05; 1.30, p = 0.04) (S3 Table). This was not statistically significant when including *APOE* as a covariate. Participants in high BP group in 1981 had higher composite cognitive scores (b = 0.46, 95%CI: 0.05; 0.87, p = 0.03), but this did not remain significant when *APOE* was included in the model (S3 Table). BP in late midlife was not associated with any of the cognitive measures (S4 Table). Those who reported high BP in very old age scored lower in semantic fluency (b = −2.89, 95%CI: −4.86; −0.91, p = 0.005), also with *APOE* in the model, but this result was not significant in immediate recall or delayed recall (S2 Table). Those reporting high BP in very old age had lower composite cognitive scores (b = −0.31, 95%CI: −0.58; −0.05, p = 0.022), also when

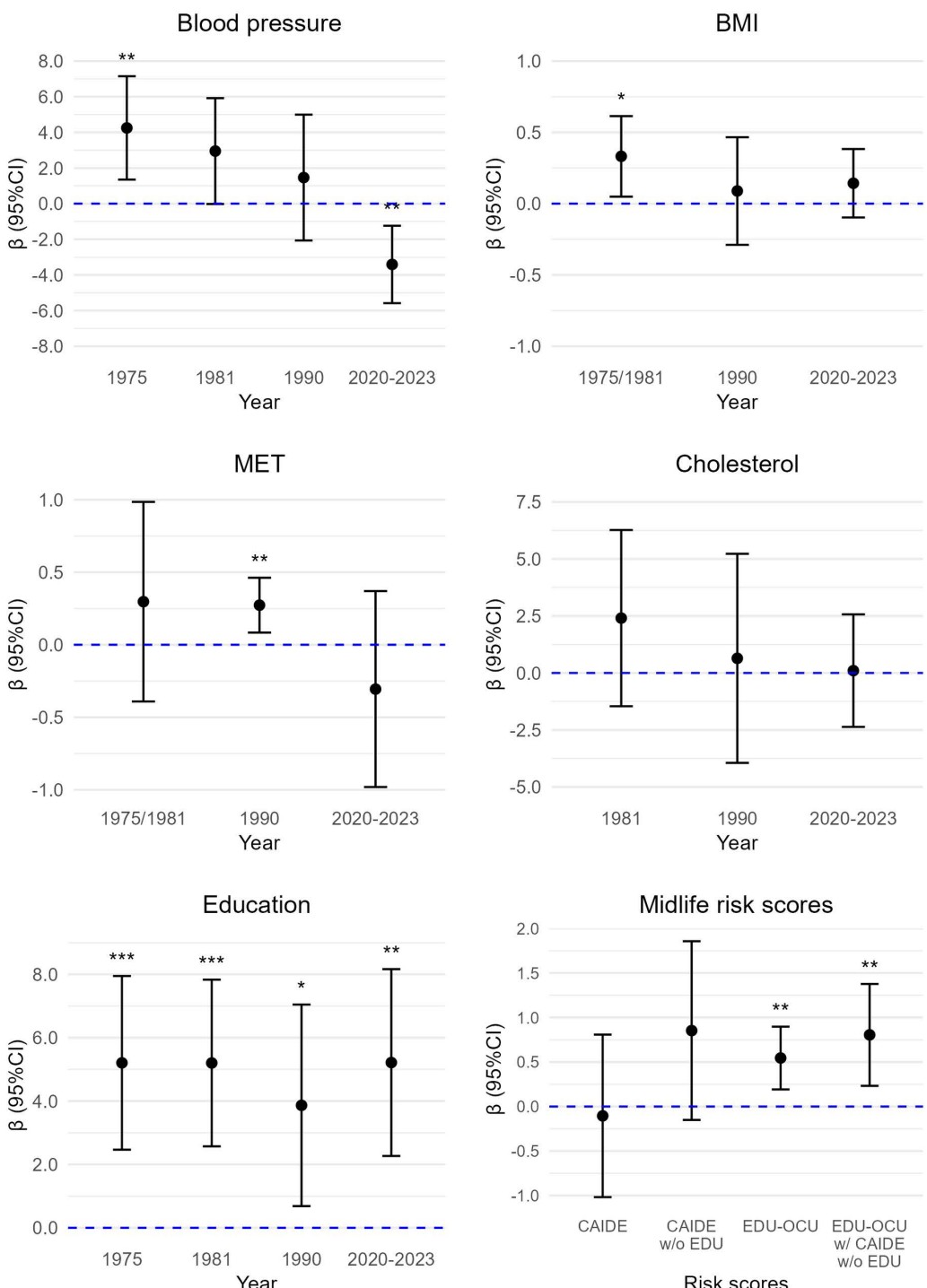

**Fig 2. Cardiovascular risk factors and midlife risk scores in association with semantic fluency.** BMI = body mass index, CAIDE = Cardiovascular Risk Factors, Aging and Dementia score, CI = confidence intervals, EDU = education, EDU-OCU = educational-occupational score, MET = metabolic equivalent hours per day, w/o = without. *p < 0.05. **p < 0.01. ***p < 0.001. Sample sizes were 37–83 (details in S2-S6 Table, model 3).

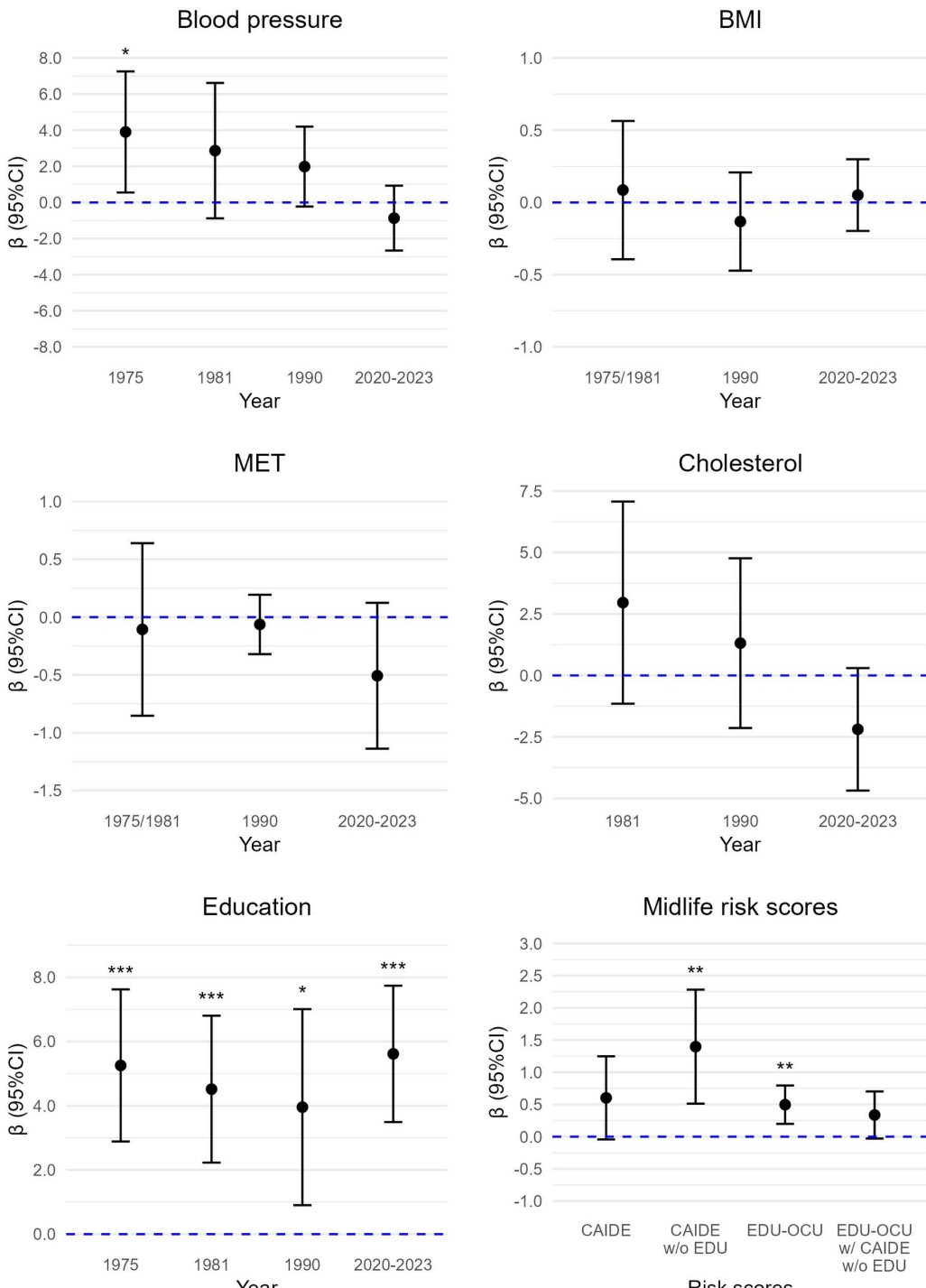

**Fig 3. Cardiovascular risk factors and midlife risk scores in association with immediate recall.** BMI = body mass index, CAIDE = Cardiovascular Risk Factors, Aging and Dementia score, CI = confidence intervals, EDU = education, EDU-OCU = educational-occupational score, MET = metabolic equivalent hours per day, w/o = without. *p < 0.05. **p < 0.01. ***p < 0.001. Sample sizes were 37–83 (details in S2-S6 Table, model 3).

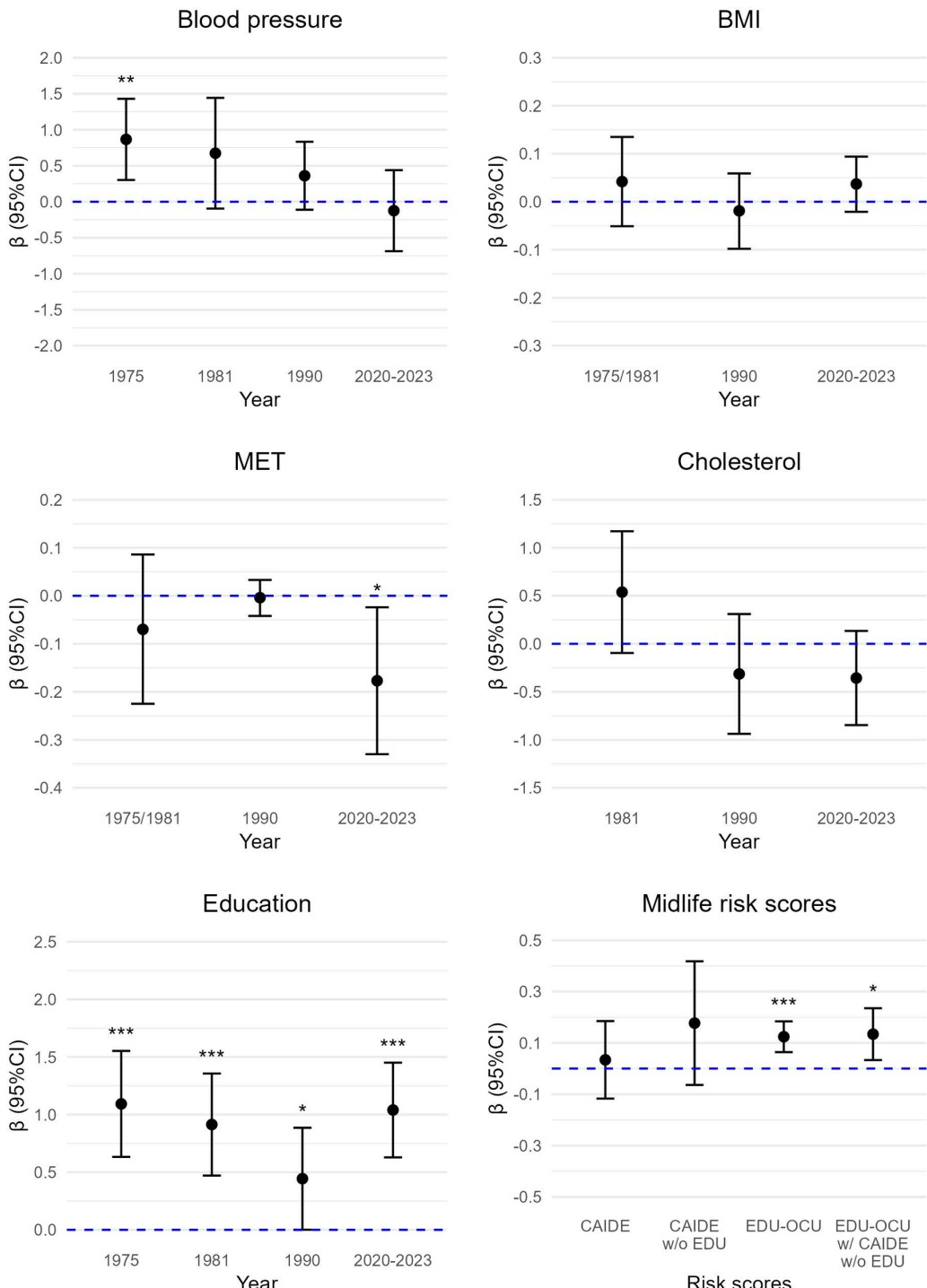

**Fig 4. Cardiovascular risk factors and midlife risk scores in association with delayed recall.** BMI = body mass index, CAIDE = Cardiovascular Risk Factors, Aging and Dementia score, CI = confidence intervals, EDU = education, EDU-OCU = educational-occupational score, MET = metabolic equivalent hours per day, w/o = without. *p < 0.05. **p < 0.01. ***p < 0.001. Sample sizes were 37–83 (details in S2-S6 Table, model 3).

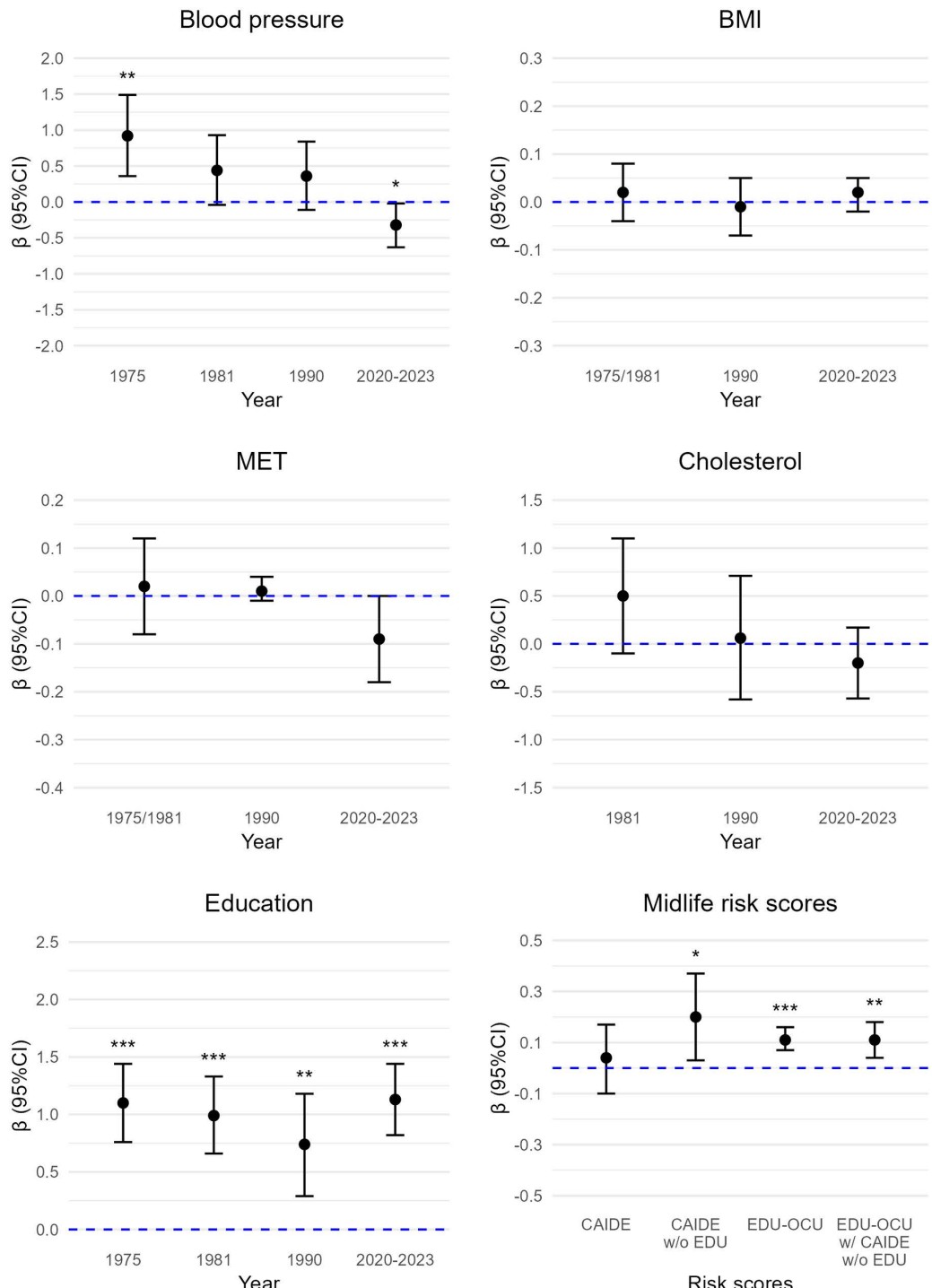

**Fig 5. Cardiovascular risk factors and midlife risk scores in association with composite cognitive score.** BMI = body mass index, CAIDE = Cardiovascular Risk Factors, Aging and Dementia score, CI = confidence intervals, EDU = education, EDU-OCU = educational-occupational score, MET = metabolic equivalent hours per day, w/o = without. *p < 0.05. **p < 0.01. ***p < 0.001. Sample sizes were 37–83 (details in S2-S6 Table, model 3).

including *APOE* in the model (S2 Table). These results for the composite cognitive score showed a medium effect size, with participants in the high BP group having 0.42 SD lower composite cognitive score than those in the normal BP group.

Total cholesterol level at any age was not significantly related to any cognitive measures (we had no measure for cholesterol in 1975) (S2-S4 Tables).

### Midlife dementia risk scores

**CAIDE.** Total CAIDE score was not significantly associated with semantic fluency, immediate recall, delayed recall, or composite cognitive score, and the results were similar with or without *APOE* as a covariate (S6 Table). Higher CAIDE score without education was associated with better semantic fluency (b = 1.05, 95%CI: 0.05; 2.05, p = 0.04), but not with *APOE* in the model. Higher CAIDE score without education was also associated with better immediate recall (b = 1.32, 95%CI: 0.44; 2.19, p = 0.004), and composite cognitive score (b = 0.21, 95%CI: 0.04; 0.37, p = 0.015) also with *APOE* in the model, but not with delayed recall, with or without *APOE* (S6 Table).

### Educational-occupational score

Educational-occupational score was positively associated with semantic fluency (b = 0.47, 95%CI: 0.13; 0.82, p = 0.008) even after adjusting for *APOE* (S6 Table). The results were similar with CAIDE risk score in the model. Educational-occupational score was also positively associated with immediate recall (b = 0.54, 95%CI: 0.31; 0.78, p < 0.001), and with delayed recall (b = 0.13, 95%CI: 0.08; 0.18, p < 0.001), and the results were similar with CAIDE score or *APOE* in the model (S6 Table). Higher educational-occupational score was also associated with higher composite cognitive score (b = 0.12, 95%CI; 0.08; 0.15, p < 0.001), also when CAIDE score and APOE were included in the models (S6 Table). The results with standardized beta coefficient for the composite cognitive score showed medium effect size indicating that one SD higher educational-occupational score corresponded to 0.44 SD higher composite cognitive score (and 0.49 SD higher composite score without CAIDE score in the model).

### Co-twin analyses

This study included 13 full twin pairs (8 MZ and 5 DZ pairs). There were only few twin pairs discordant for CV risk factors, in midlife or very old age (S7 Table). Detailed description of discordancy can be found in the Supplement (S1 Text).

### Drop-out analyses

Participants who completed both the telephone interview and questionnaire at age 90 had significantly higher educational-occupational scores (p < 0.001), lower total CAIDE scores (p = 0.006), and lower CAIDE score without education (p = 0.029) than those who completed only the questionnaire. They also had higher educational-occupational scores (p < 0.001) and lower CAIDE scores (p = 0.013) than those who did not participate in neither. Participants were less likely to have high BP in 1990 than those with only questionnaires (p = 0.012). They also had higher education levels in 1975, 1981, and at age 90 (all p < 0.001). See Supplement for detailed results (S1 Text, S9-S10 Table). Results from the IPW analyses are presented in the Supplement (Tables S11–S15). The findings remained consistent with the original analyses. The main difference observed in the IPW analyses was that the educational-occupational score was no longer significantly associated with semantic fluency, although it remained significantly associated with immediate recall, delayed recall, and the composite cognitive score.

## Discussion

In line with previous studies on younger cohorts [7,22] and in 90-year olds [23], higher education was associated with better cognitive functioning at 90 years, suggesting that education is associated with better cognition across the life span.

Effect sizes for the composite cognitive score ranged from medium to large: participants with 7–11 years of education had 0.41 SD higher score than those with 6 years or less, and those with 12 or more years of education had even 1.45 SD higher score. Correspondingly, the educational-occupational score, which addresses midlife, showed a positive association with cognition, with one SD higher educational-occupational score corresponding to 0.44 SD higher composite cognitive score. The educational-occupational score can be considered as a proxy of cognitive reserve, reflecting both lifelong exposure to intellectually demanding environments and possibly also young age cognitive ability [12,24,25]. The educational-occupational score might be more directly linked to intellectual engagement and resilience against age-related cognitive decline, compared to CAIDE which emphasizes vascular health. It should be noted that higher childhood intelligence has been associated with both higher old age cognition [26] and lower CV disease risk [27]. We did not have a direct measure for childhood intelligence, which is a limitation in this study. Nevertheless, higher educational attainment is listed as one of the most influential protective factor against dementia in the latest Lancet commission report on dementia risk factors [7].

Also, the individual CV risk factors in our study were included in the latest Lancet commission report [7]. While better overall CV health has been associated with decreased dementia risk [28], studies have found that modifiable risk factors impact cognition differently across life stages [3–6]. Some studies looking at CV risk factors measured in old age have reported cross-sectional associations between higher BMI and hypertension and better cognitive function or lower risk of dementia [6,9,29]. In our study, midlife or late-life PA, BMI, cholesterol, and CAIDE scores showed no link to cognition at age 90, suggesting CV risk factors may play a different role in cognitive health for those reaching very old age. The CAIDE score was originally developed to predict 20-year dementia risk, and it might not be appropriate for predicting cognitive function at age 90, or outcome based on cognitive tests. A sample consisting of individuals who have lived past 90 years old is also likely to introduce survivor bias in the study, resulting in unexpected associations of the midlife and old age risk factors. It is also possible that cognitive risk factors after 90 differ from those affecting younger older adults (60+), as previous studies suggest [23].

Participants who belonged in the high BP group in 1975 had better cognition in very old age, showing quite large effect sizes with participants with high BP having over one SD higher composite cognitive score than those with normal BP. Midlife hypertension is one of the most common dementia risk factors, associated with nearly 60% increased relative risk of developing all cause dementia [24]. Mid-to-late-life antihypertensive medication use has been linked to lower dementia risk, while untreated hypertension raises risk [30,31]. In our study, all but one participant who reported using antihypertensive medications in midlife belonged in the high BP group. The earlier start of antihypertensive medications could explain the unexpected association between high BP in midlife and better cognition in very old age. Considering the small sample size and low number of people in the high BP groups, it is possible these findings reflect reverse causality or selection bias. In 1970s Finland, many were unaware of having high BP [32], and some participants may have been misclassified as having normal BP, while those on medication might represent a group with better access to services.

This study's strengths include a 48-year follow-up and a population-based sample with a nearly equal gender balance. There are several limitations in this study which warrant cautious interpretation of our findings. It is important to consider the small sample size and limited statistical power of the regression analyses, which increases the risk of failing to detect true effects. Additionally, including a high number of covariates relative to the sample size can lead to unstable estimates and inflated standard errors, which may limit the reliability and generalizability of the findings. The healthy survivor effect can also influence our results, as individuals who survive to older ages, especially above 90-years, usually represent a subset of a population with better health profiles. This may also generate spurious associations, including reverse causations. No formal correction for multiple testing was applied due to the exploratory nature of the study and the limited sample size, increasing the potential for false positive findings. However, findings were interpreted with caution, considering consistency across cognitive measures and effect sizes rather than relying solely on p-values. Heterogeneity in the risk profiles could influence the findings, as 90-year-olds tend to have a high prevalence of risk factors (e.g., high BP,

low PA), diluting differences between individuals [33]. Lastly, telephone based cognitive assessment for the oldest old likely includes more measurement error related to sensory limitations, distraction and comprehension issues compared to younger old adults. A floor effect was observed in delayed recall, but this was accounted for in the statistical analyses. These limitations highlight the need for future research with larger, more representative samples and optimized study designs that enable more precise estimation of effects and reduce the risk of spurious findings.

Participants had higher education and lower CAIDE scores than those who only completed the questionnaire at age 90 or did not participate. Plans to compare co-twins with differing risk factors were not conducted due to high concordance in risk factors and small number of twin pairs. Prior studies in Finland have shown good agreement between self-reports, medical records, and in person measures [34–36].

## Conclusion

Education remains as a protective factor for cognition in above 90-year-olds. However, given that educational attainment may reflect early-life cognitive ability and socio-economic status, which were not directly measured in this study, this result must be interpreted cautiously. Our results suggest that CV risk factors are of at most modest importance, but due to sample size and selection effects, more research is needed to confirm this. Future research should focus on investigating the contributions of early-life cognitive ability and educational attainment using longitudinal life course data. Pooling data across multiple population-based studies would improve statistical power and allow for better understanding of population heterogeneity in these associations.

## Supporting information

**S1 Text. Detailed description of variables and drop-out analyses.**
(DOCX)

**S1 Table. Alcohol use in a month.**
(DOCX)

**S2 Table. Linear regression analysis results for lifestyle factors at 90 years old.**
(DOCX)

**S3 Table. Linear regression analysis results for lifestyle factors in 1981.**
(DOCX)

**S4 Table. Linear regression analysis results for lifestyle factors in 1990.**
(DOCX)

**S5 Table. Linear regression analysis results for lifestyle factors in 1975.**
(DOCX)

**S6 Table. Linear regression analysis results for CAIDE and educational-occupational score.**
(DOCX)

**S7 Table. Discordant twin pairs.**
(DOCX)

**S8 Table. Descriptive statistics for those who participated in telephone interviews and questionnaires in 90 years old, and those who only participated in questionnaire at age 90.**
(DOCX)

**S9 Table. Two-sample t-test results.**
(DOCX)

**S10 Table. Desing based F-test results.**
(DOCX)

**S11 Table. Inverse probability weighted linear regression analysis results for lifestyle factors at 90 years old.**
(DOCX)

**S12 Table. Inverse probability weighted linear regression analysis results for lifestyle factors in 1981.**
(DOCX)

**S13 Table. Inverse probability weighted linear regression analysis results for lifestyle factors in 1990.**
(DOCX)

**S14 Table. Inverse probability weighted linear regression analysis results for lifestyle factors in 1975.**
(DOCX)

**S15 Table. Inverse probability weighted linear regression analysis results for CAIDE and educational-occupational score.**
(DOCX)

## Acknowledgments

We would like to thank all the twins who participated in the study.

## Author contributions

**Conceptualization:** Eero Vuoksimaa.

**Data curation:** Teemu Palviainen, Mia Urjansson, Eero Vuoksimaa.

**Formal analysis:** Anni Varjonen, Paula Iso-Markku, Jaakko Kaprio, Eero Vuoksimaa.

**Funding acquisition:** Jaakko Kaprio, Eero Vuoksimaa.

**Project administration:** Mia Urjansson, Eero Vuoksimaa.

**Resources:** Jaakko Kaprio, Eero Vuoksimaa.

**Supervision:** Toni Saari, Eero Vuoksimaa.

**Visualization:** Anni Varjonen.

**Writing – original draft:** Anni Varjonen.

**Writing – review & editing:** Toni Saari, Sari Aaltonen, Teemu Palviainen, Paula Iso-Markku, Jaakko Kaprio, Eero Vuoksimaa.

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
