## [Decision Letter · Decision Letter 0]

15 May 2025

PONE-D-25-11838Midlife and old-age cardiovascular risk factors, educational attainment, and cognition at 90-years – population-based study with 48-years of follow-upPLOS ONE

Dear Dr. Varjonen,

Thank you for submitting your manuscript to PLOS ONE. After careful consideration, we feel that it has merit but does not fully meet PLOS ONE’s publication criteria as it currently stands. Therefore, we invite you to submit a revised version of the manuscript that addresses the points raised during the review process. Please submit your revised manuscript by Jun 29 2025 11:59PM. If you will need more time than this to complete your revisions, please reply to this message or contact the journal office at plosone@plos.org . Please include the following items when submitting your revised manuscript:

A rebuttal letter that responds to each point raised by the **academic editor and reviewers** . You should upload this letter as a separate file labeled 'Response to Reviewers'.A marked-up copy of your manuscript that highlights changes made to the original version. You should upload this as a separate file labeled 'Revised Manuscript with Track Changes'.An unmarked version of your revised paper without tracked changes. You should upload this as a separate file labeled 'Manuscript'.STROBE checklist with page numbersFlow diagram (redesigned)

We look forward to receiving your revised manuscript.

Kind regards,

Gareth Hagger-Johnson

Academic Editor

PLOS ONE

Journal Requirements:

Additional Editor Comments:

This could potentially be suitable for publication in PLOS One, but will require substantial revision and some additional analyses.

In addition to addressing comments and concerns raised by the reviewers, please address my comments below.

The research questions and analytic plan do not appear to have been pre-registered (e.g. Open Science Framework or similar). Given the number of statistical tests, this increases the likelihood of chance and unstable findings.

Is there anywhere else the questions and plan were stated in advance (e.g. data sharing applications, ethics forms)? Include if possible.

There is no mention of childhood intelligence yet a substantial literature on the long-reaching consequences. Educational attainment is not the same thing, particularly for older cohorts and older women, whose years of education would be influenced by sociological factors not just childhood intelligence. At the very least, please acknolwedge this literature and the limitation that you were not able to consider this in your analysis.

I was suprising not to see use of inverse probability weights to address non-random attrition and drop-out (and reduce bias). You could upweight participants similar to non-responders and dropouts, to reduce possible bias (or at least evaluate differences vs. unweighted analysis). This is fairly common an approach.

Please construct a composite of global/general function - all three tests, for comparison. I would have done global function as the main analysis, then specific tests as secondary or even supplementary/exploratory analysis.

The role of healthy survivor effects and very small sample size (low statistical power) are mentioned only briefly but are fundamental limitations. Much more emphasis and reflection on this is needed, particularly in the discussion section.

Healthy survivor effects could create spurious and associations in the opposite direction to the causal effects.

P2 ABSTRACT. Is "Cardiovascular Risk Factors, Aging, and Dementia (CAIDE)" and "dementia risk factors" the same thing? Reword for clarity to avoid confusion.

P2L36 Do you mean cognitive aging or aging?

P2L40 "substantial proportion" and "high prevalence" reads akwardly. Can you report prevalence for both with the same term?

P3L54 "midlife and old age" is confusing - state age range more simply. Then there can be no doubt about the meaning of each term.

P3L56 It reads strangely that you are using "previously validated" dementia risk scores yet your question concerns their validity in a population they were not validated for. If the aim of the study is to validate them on an older age, be more clear about this. If the assumption is made that using risk scores validated at midlife will suffice on old age adults, that sounds quite weak, so needs rewriting for clarity and strength.

P3L58 Hypothesized - but not pre-registered anywhere?

P3L61 Missing statement "..have a reversed association with cognition ON THE BASIS THAT...."

P3L65 I do not understand this: "irrespective of co-twin's vital status or participation". Do you mean any surviving twin was eligible? Please use the STROBE language (eligible, recruited, analysed etc.) and add flow diagram.

P4 The NONAGINTA term suddenly appears from nowhere and is quite jarring. It needs a brief introduction and rewording e.g. to make it clear that this is a nested/sub study of a larger cohort.

P4L73 Why is the 1990 cohort so much smaller than the others?

P4L81 Here and elsewhere, please use age range observed and year(s) observed. Readers not involved in the study need to quickly understand how old and when data were collected.

P4L89 Be careful creating binary variables, you lose variation and results are more likely to drift to null. See commentary on dichotomania (https://www.fharrell.com/post/errmed/)

P5L92 Language - "had used" not "have used". "none in 1981" not "while none in 1981"

P5L95 If you must used binary variables, simpler here to say normal/low and high rather than "One person...."

P5L103 Why frequency of alcohol use but not quantity or hazardous pattern?

P5L107 Careful with the term "this study" which can be confusing when writing up specific questions in samples nested in wider cohorts. Some readers will get confused about what "study" this refers to, particularly if you have just mentioned three sets of questionnaires over many years. Consider "For our analysis" or similar.

P6L114 I don't understand this: "as opposed to original CAIDE with in-person measurements" and so might not readers.

P6L113 I appreciate the need to reference previously published cut-points [8, 12, 8] but can you briefly state some additional information so that we don't have to look up and read these papers separately? It's a bit too brief currently.

P6L145 I appreciate the drop-out analysis, but many readers (including me) are more familiar with inverse probability weighting to evaluate drop-out bias i.e. logistic regression of drop-out and end of follow-up based on socio-demographic and other predictors at baseline, save probability, take inverse, use inverse to upweight responders who look most similar to dropouts. Compare results side by side - materially different?

P6L153 Returning a questionnaire is widely considered to represent informed consent, particulalry in an existing study where participants are used to completing them. Why would that differ today - can you briefly explain (I wasn't sure why newer legislation needs to be mentioned if it was ethical and legal).

P8 Table 1 - Why the sudden drop in 1990 to N = 53 (see earlier point - inclusion criteria at this phase not clear, needs STROBE detail). Why APOE-e4 different proportions for men/women?

P9L179 Healthy survivor effects - very few smokers remained at older ages.

P12L257 The drop-out analysis is informative but descriptive - it is does not evaluate the impact of bias on substantive results, whereas IPW might to some extent.

P12L267 Nothing mentioned on childhood IQ which could drive CVD risk, cognitive aging, dementia risk across the lifecourse. Studies should be cited on this missing antecedent variable.

P13L277 Be clearer here - are you saying hypertension and higher BMI were associated with better cognitive function (cross-sectionally), less cognitive decline. Longitudinally? Were these impacted potentially by reverse causaal, healthy survivor, dropout effects? Are there any administrative data / whole population studies which can clarify or compare?

P13L282 Suggest "participants" not "people" here.

P13L283 Rewrite for clarity: "...is one of the most prevalent dementia risk factors, with nearly 60% increased risk...". I do not understand this sentence. Do you mean being hypertensive at midlife is the most commonly ocurring risk factor, and that aditionally it increases relative risk by 60%, or 60% of dementia is attributable to hypertension at midlife, or something else? Rewrite so that it cannot be misunderstood.

P14L294 This section feels a little rushed e.g. "should be considered when interpreting results". Why? Be specific. Lean into your main limitations - low statistical power, healthy survivor effects, multiple testing, lots of predictors, lots of heterogeneity. Position these with clear next steps for future researchers to design precise research questions and better study designs.

P14L300 This one sentence on 'selected group' understates the problem - it could be a whole paragraph in my view. Also - IPW might fundamentally offer different and more interesting results, even if to show how different the results are when explicitly modelling dropout characteristics using weights.

P14L305 Careful here - you are making policy statements (access to education) but childhood IQ was not measured. In older cohorts, number of years of education would be correlated with childhood IQ but societal factors might drive number of years, particularly for women. You cannot disentangle childhood intelligence from educational attainment easily (and not at all in this dataset), so you should not be making statements about educational policy and resource allocation. Reword as possible mechanisms for future study in longitudinal (ideally life course longitudinal) studies. Also, given low power, you should make the suggestions that studies from several populations should be pooled together, to improve power and study heterogeneity of effects (e.g. individual participant meta-analysis).

Figure 1. This needs redesigning slightly - the arrows were confusing me. NONAGINTA was created in 2020/23 but the arrows moving to the right refer to it from 1975 onwards, higher up. Please design so that it is clearer at first glance, that NONAGINTA is nested within a wider cohort. See STROBE guidelines and flow diagram examples. Again, I wasn't clear why the 1990 cohort was so much smaller - is this by design? What is the reason?

Reviewers' comments:

Reviewer's Responses to Questions

**Comments to the Author**

1. Is the manuscript technically sound, and do the data support the conclusions?

Reviewer #1: Partly

Reviewer #2: Yes

2. Has the statistical analysis been performed appropriately and rigorously? 

Reviewer #1: N/A

Reviewer #2: Yes

3. Have the authors made all data underlying the findings in their manuscript fully available?

Reviewer #1: Yes

Reviewer #2: Yes

4. Is the manuscript presented in an intelligible fashion and written in standard English?

Reviewer #1: Yes

Reviewer #2: Yes

5. Review Comments to the Author

Reviewer #1: Dear Authors,

Thank you for the opportunity to review your manuscript entitled “Midlife and old-age cardiovascular risk factors, educational attainment, and cognition at 90-years – population-based study with 48-years of follow-up.”

I kindly commend you on an ambitious and important study that uses a rich, population-based twin cohort to investigate the long-term impact of education and cardiovascular health on cognition in the oldest-old. The extensive follow-up and the integration of both midlife and late-life risk factors add considerable value to the field of cognitive aging. Please find below my feedback.

I believe these revisions will substantially strengthen the manuscript. Thank you again for your thoughtful and important contribution to the literature on aging and cognitive health.

With best regards,

The Reviewer

-----

Major Points

1. Sample Size and Statistical Power

• The final sample size (N = 96) is modest, especially for subgroup comparisons (e.g., BP categories), and likely underpowered for some outcomes such as delayed recall.

• Please comment on the statistical power for the primary outcomes. Consider reporting effect sizes (e.g., Cohen’s d or beta coefficients with 95% CI) consistently throughout to contextualize findings.

2. Reverse Causation and Selection Bias

• The finding that midlife hypertension is associated with better cognitive outcomes is contrary to existing literature and may reflect reverse causality or selection bias.

• Discuss this thoroughly and consider stratified analyses (e.g., by antihypertensive medication use).

• Consider controlling for or reporting on duration of medication use.

• Acknowledge potential survivor bias more explicitly in both the results and discussion.

3. Interpretation of CAIDE Score

• The CAIDE score did not predict cognitive outcomes, despite being a validated dementia risk tool. This discrepancy is insufficiently addressed.

• Discuss potential reasons why CAIDE performed poorly.

• Clarify how CAIDE scores were calculated from self-report.

• Consider additional analyses excluding the education component or including APOE status.

4. Educational-Occupational Score

• This composite measure consistently predicted cognition, outperforming CAIDE.

• Clarify what time period the work-related questions refer to. Discuss the extent to which this score may reflect cognitive reserve, and how it relates to lifelong intellectual engagement.

5. Delayed Recall Floor Effects

• The average delayed recall score (2.2 out of 10) suggests a potential floor effect.

• Consider alternative modeling strategies (e.g., zero-inflated or transformed models).

• Discuss limitations of the telephone-based assessment in very old adults.

Minor Comments

• Abstract: Consider rephrasing “no consistent associations” with more precise language.

• Figure captions: Add sample sizes and clarify whether multiple comparisons were corrected.

• Define what constitutes “high” vs. “normal” BP based on Finnish guidelines in the 1970s–1980s.

• Specify how missing data were handled (e.g., imputation, complete-case analysis).

• Clarify APOE genotyping procedures (e.g., DNA source, platform).

Questions for the Authors

1. Can you elaborate on how education was treated longitudinally—was the highest value used across waves?

2. Could the paradoxical effect of midlife hypertension reflect early treatment or better access to care? Have you considered stratifying analyses based on antihypertensive use?

3. Do you view the educational-occupational score as a proxy for cognitive reserve? If so, how does this fit with the null CAIDE findings?

4. Do attrition analyses suggest meaningful differences in baseline risk factors between completers and non-completers?

5. How might findings generalize to more diverse, less homogeneous populations?

Reviewer #2: This study examined the association between cognitive function and cardiovascular risk factors. 96 participants were followed up for an average of 46 years. High numbers of statistical test were performed and the main result showed that high education was associated with better cognitive functioning at 90 years of age. Some other associations were also found but these remained a bit inconsistent. Interestingly, CAIDE score was not associated with cognitive function.

The long follow-up time is the greatest strength of this study. However, the number of participants is quite low, N=96. This is a significant concern that brings a lot of uncertainty to the interpretation of the results. Low number of participants and high number of statistical tests raises the concern of multiple testing problem.

Furthermore, I have some other comments.

1. What were the shortest and the longest follow-up time?

2. In covariates, blood pressure and total cholesterol were used as binary variables where high and normal levels were evaluated based on self-reported questionnaire. This raises some concerns because, for example, very low blood pressure has been reported to be associated with poor cognitive function or increased dementia risk. The self-reported high or normal levels may increase the risk of bias because the cut-off values for the binary variables are not known at all. It would be clearer to state in the actual manuscript that the covariates are not based on actual measurements.

3. Please address the potential role of multiple testing on the observed associations.

4. There are no details on how many of participants had dementia. How are the risk factors associated with dementia risk?

6. PLOS authors have the option to publish the peer review history of their article (what does this mean? ). If published, this will include your full peer review and any attached files.

**Do you want your identity to be public for this peer review?** For information about this choice, including consent withdrawal, please see our Privacy Policy .

Reviewer #1: **Yes: ** Miray Budak

Reviewer #2: **Yes: ** Juuso O. Hakala

---

## [Author Response · Author response to Decision Letter 1]

9 Aug 2025

Responses to reviewers

We thank the reviewers and the editor for their insightful comments which have helped improve the manuscript. The changes in the manuscript are marked with Track Changes and by page number (referring to the document with track changes) in this response letter. The responses in this document are listed in the following order: comments by the editor, comments by reviewer #1, and comments by reviewer #2.

EDITORS COMMENTS:

1. If there are ethical or legal restrictions on sharing a de-identified data set, please explain them in detail (e.g., data contain potentially identifying or sensitive patient information, data are owned by a third-party organization, etc.) and who has imposed them (e.g., a Research Ethics Committee or Institutional Review Board, etc.). Please also provide contact information for a data access committee, ethics committee, or other institutional body to which data requests may be sent.

Response: Because of the consent given by the participants at the time of data collection, and the easier identification of twin data and sensitive nature of the questionnaire data, the data is not publicly available. Data are available through the Institute for Molecular Medicine Finland (FIMM) Data Access Committee (DAC), who have IRB/ethics approval and an institutionally approved study plan. This information and the contact information of the FIMM DAC (fimm-dac@helsinki.fi) is provided in the data availability statement in the manuscript.

2. If there are no restrictions, please upload the minimal anonymized data set necessary to replicate your study findings to a stable, public repository and provide us with the relevant URLs, DOIs, or accession numbers. Please see http://www.bmj.com/content/340/bmj.c181.long for guidelines on how to de-identify and prepare clinical data for publication. For a list of recommended repositories, please see https://journals.plos.org/plosone/s/recommended-repositories. You also have the option of uploading the data as Supporting Information files, but we would recommend depositing data directly to a data repository if possible.

Response: The restrictions for data availability are mentioned in the data availability statement, and contact information is provided.

Response: Captions for supporting information files have been added at the end of the manuscript file, and all in-text citations have been updated accordingly.

4. The research questions and analytic plan do not appear to have been pre-registered (e.g. Open Science Framework or similar). Given the number of statistical tests, this increases the likelihood of chance and unstable findings. Is there anywhere else the questions and plan were stated in advance (e.g. data sharing applications, ethics forms)? Include if possible.

Response: Unfortunately, the research plan was not pre-registered. A section was added to the manuscript (see page 18, discussion) mentioning the issue of multiple testing and how it could affect the observed associations.

5. There is no mention of childhood intelligence yet a substantial literature on the long-reaching consequences. Educational attainment is not the same thing, particularly for older cohorts and older women, whose years of education would be influenced by sociological factors not just childhood intelligence. At the very least, please acknowledge this literature and the limitation that you were not able to consider this in your analysis.

Response: The first paragraph of the discussion section (see page 16) was almost completely re-written to better describe how childhood intelligence and its long-reaching effects on cognitive function in old age relate to implication of our study results.

6. I was surprising not to see use of inverse probability weights to address non-random attrition and drop-out (and reduce bias). You could upweight participants similar to non-responders and dropouts, to reduce possible bias (or at least evaluate differences vs. unweighted analysis). This is fairly common an approach.

Response: We agree with this observation, and we have now conducted the inverse probability weighted analyses, and these are included in the revised manuscript. We used logistic regression to predict staying in the study based on sex and educational-occupational score, which differed significantly between participants and non-responders. The educational-occupational score also captures more information on socio-economic status than education alone. CAIDE score was not used due to the substantial reduction of the sample size for this variable. The regression analyses were then conducted again using stabilized inverse probability weights. The findings remained consistent with the original analyses and accordingly, these analyses did not change the conclusions of this study.

Tables with results have been added to the supplement (tables S17- S21), and following description of this analysis method was included in the main manuscript (page 8): “We also conducted an inverse probability weighting (IPW) analysis to adjust for potential bias due to non-random dropout. Logistic regression was used to estimate the probability of remaining in the study, with predictors including the educational-occupational score (which differed significantly between participants and non-participants), sex, and family relatedness (accounted for via clustering). Stabilized inverse probability weights were then applied in the regression models.”

Following text was added to the Result section (page 16): “Results from the IPW analyses are presented in the Supplement (Tables S17–S21). The findings remained consistent with the original analyses. The main difference observed in the IPW analyses was that the educational-occupational score was no longer significantly associated with semantic fluency, although it remained significantly associated with immediate recall, delayed recall, and the composite cognitive score.”

7. Please construct a composite of global/general function - all three tests, for comparison. I would have done global function as the main analysis, then specific tests as secondary or even supplementary/exploratory analysis.

Response: We agree with this suggestion, and we have constructed a composite cognitive score (mean of z-scores from each individual cognitive test) and used this in the revised manuscript. This outcome variable was included in all analyses, and the results added to the regression tables. A new figure was included for the composite cognitive score (Fig 5). The findings are mainly in line with the original findings. This additional analysis was incorporated in the manuscript text throughout.

8. The role of healthy survivor effects and very small sample size (low statistical power) are mentioned only briefly but are fundamental limitations. Much more emphasis and reflection on this is needed, particularly in the discussion section. Healthy survivor effects could create spurious and associations in the opposite direction to the causal effects.

Response: The limitations section in the Discussion has been almost completely re-written (pages 17-18), discussing the issues posed by the small sample size and the possible influence of the healthy survivor bias and reverse causation. In addition, more caution has been added to discussion of our results throughout the Discussion section.

P2 ABSTRACT. Is "Cardiovascular Risk Factors, Aging, and Dementia (CAIDE)" and "dementia risk factors" the same thing? Reword for clarity to avoid confusion.

Response: Added “individual” cardiovascular risk factors to more clearly make a distinction that we investigated the risk factors individually and together as the CAIDE score.

P2L36 Do you mean cognitive aging or aging?

Response: Corrected to “cognitive aging”

P2L40 "substantial proportion" and "high prevalence" reads awkwardly. Can you report prevalence for both with the same term?

Response: Removed “substantial proportion”

P3L54 "midlife and old age" is confusing - state age range more simply. Then there can be no doubt about the meaning of each term.

Response: Modified to include age range and refer to “very old age” rather than “old age.” The manuscript, page 3, now reads: “Our aim was to use a population-based sample of twins with up to 48 years of follow-up to investigate if CV risk factors measured in midlife (age range 42-51) and very old age (age range 90-98), including BMI, blood pressure (BP), cholesterol, and physical activity (PA) predict cognition at nonagenarian age.”

P3L56 It reads strangely that you are using "previously validated" dementia risk scores yet your question concerns their validity in a population they were not validated for. If the aim of the study is to validate them on an older age, be more clear about this. If the assumption is made that using risk scores validated at midlife will suffice on old age adults, that sounds quite weak, so needs rewriting for clarity and strength.

Response: This sentence was added for clarity (page 3): “While these scores were originally developed and validated to predict late-life dementia based on midlife risk factors, our aim here was to explore whether these midlife profiles remain predictive of cognitive outcomes at very old age.”

P3L58 Hypothesized - but not pre-registered anywhere?

Response: The study was not pre-registered.

P3L61 Missing statement "..have a reversed association with cognition ON THE BASIS THAT...."

Response: Added sentence (page 4): “…on the basis that in advanced stages of dementia, certain risk factors such as low BMI and hypertension may no longer indicate risk, but rather reflect the severity or progression of the disease itself [13].”

P3L65 I do not understand this: "irrespective of co-twin's vital status or participation". Do you mean any surviving twin was eligible? Please use the STROBE language (eligible, recruited, analyzed etc.) and add flow diagram.

Response: Added sentence (page 4): “All eligible surviving twins were recruited for the study, regardless of whether their co-twin was deceased or did not participate.” The data collection and waves of questionnaires are described in a flow diagram (Fig 1) that has been modified according to the comments in this review for more clarity.

P4 The NONAGINTA term suddenly appears from nowhere and is quite jarring. It needs a brief introduction and rewording e.g. to make it clear that this is a nested/sub study of a larger cohort.

Response: Sentence was modified to include better wording leading into the NONAGINTA term (page 4): “…and participated in a sub study focusing on twins turning 90-years-old, referred to as the NONAGINTA –study.”

P4L73 Why is the 1990 cohort so much smaller than the others?

Response: For the 1975 and 1981 questionnaires, all twins born prior to 1958 were invited for the mail-in questionnaire. In 1990, only twins born between 1930-1958 were invited, excluding those born before 1930, making the cohort smaller. A sentence was added/modified to clarify this in the methods section (page 4): “We also used earlier FTC data that were collected through postal questionnaires with a baseline in 1975 (all twins), and follow-ups in 1981 (all twins) and 1990 (only those born 1930 or later were invited).”

P4L81 Here and elsewhere, please use age range observed, and year(s) observed. Readers not involved in the study need to quickly understand how old and when data were collected.

Response: The age range was added to the text.

P4L89 Be careful creating binary variables, you lose variation and results are more likely to drift to null. See commentary on dichotomania (https://www.fharrell.com/post/errmed/)

Response: For blood pressure and cholesterol variables, which were based on self-reports and categorical variables, there were unfortunately too few participants in each group, and binary variables were the only option. Blood pressure and cholesterol measures were not available for the entire cohort at the earlier time-points.

P5L92 Language - "had used" not "have used". "none in 1981" not "while none in 1981"

Response: Corrected

P5L95 If you must used binary variables, simpler here to say normal/low and high rather than "One person...."

Response: Corrected

P5L103 Why frequency of alcohol use but not quantity or hazardous pattern?

Response: Majority of the participants reported frequency of alcohol use to be very low or none, which is common in the 90-year-old cohort. Based on this we determined frequency to be sufficient on its own to describe alcohol use in this cohort.

P5L107 Careful with the term "this study" which can be confusing when writing up specific questions in samples nested in wider cohorts. Some readers will get confused about what "study" this refers to, particularly if you have just mentioned three sets of questionnaires over many years. Consider "For our analysis" or similar.

Response: Changed to “for our analysis” everywhere where relevant.

P6L114 I don't understand this: "as opposed to original CAIDE with in-person measurements" and so might not readers.

Response: More detailed explanation was added to improve clarity (page 6): “CAIDE score was based on scores calculated in [8]. The score was determined from participants’ self-reports in 1975, 1981, and 1990 questionnaires, including BP, BMI (from height and weight), cholesterol levels, and exercise frequency. When data were missing at one time point, values from an alternate year or later questionnaire (e.g., 1990 for cholesterol) were used. The classification thresholds and scoring followed those of the original CAIDE model, with minor adjustments to accommodate available data. Original CAIDE was developed using in-person measurements [11].”

P6L113 I appreciate the need to reference previously published cut-points [8, 12, 8] but can you briefly state some additional information so that we don't have to look up and read these papers separately? It's a bit too brief currently.

Response: More detail was added to these paragraphs, especially information about the educational-occupational score (see pages 6-7). Revised text reads: “The educational-occupational score was based on the scores calculated in [12]. It consisted of the following self-reported variables from the 1975 (mean age 45) and 1981 (mean age 52) postal questionnaires: age, years of education, work status (not working, homemaker, or working/ studying), complexity of work (very monotonous, somewhat monotonous/variable, or very variable), physical loading of work (heavy manual labor, manual labor with lifting, light manual labor, and nonmanual work) and work environment (mainly outdoors or both indoors and outdoors, vs. indoors only) [8]. The questions for these variables and their categorization are explained in detail in [8].”

P6L145 I appreciate the drop-out analysis, but many readers (including me) are more familiar with inverse probability weighting to evaluate drop-out bias i.e. logistic regression of drop-out and end of follow-up based on socio-demographic and other predictors at baseline, save probability, take inverse, use inverse to upweight responders who look most similar to dropouts. Compare results side by side - materially different?

Response: Inverse probability weighting was conducted to evaluate drop-out bias. Detailed explanation of this can be found in the response for editor comment #6.

P6L153 Returning a questionnaire is widely considered to represent informed consent, particularly in an existing study where participants are used to completing them. Why would that differ today - can you briefly explain (I wasn't sure why newer legislation needs to be mentioned if it was ethical and legal).

Response: Since there is a separate mention for the NONAGINTA study approval and written informed consent, we referred to Finnish legislation to clarify that the framework under which the study began was ethical an

---

## [Decision Letter · Decision Letter 1]

15 Aug 2025

Midlife and old-age cardiovascular risk factors, educational attainment, and cognition at 90-years – population-based study with 48-years of follow-up

PONE-D-25-11838R1

Dear Dr. Varjonen,

We’re pleased to inform you that your manuscript has been judged scientifically suitable for publication and will be formally accepted for publication once it meets all outstanding technical requirements.

Kind regards,

Gareth Hagger-Johnson

Academic Editor

PLOS ONE

Reviewers' comments:

Reviewer's Responses to Questions

**Comments to the Author**

1. If the authors have adequately addressed your comments raised in a previous round of review and you feel that this manuscript is now acceptable for publication, you may indicate that here to bypass the “Comments to the Author” section, enter your conflict of interest statement in the “Confidential to Editor” section, and submit your "Accept" recommendation.

Reviewer #2: All comments have been addressed

2. Is the manuscript technically sound, and do the data support the conclusions?

Reviewer #2: Yes

3. Has the statistical analysis been performed appropriately and rigorously? 

Reviewer #2: Yes

4. Have the authors made all data underlying the findings in their manuscript fully available?

Reviewer #2: No

5. Is the manuscript presented in an intelligible fashion and written in standard English?

Reviewer #2: Yes

6. Review Comments to the Author

Reviewer #2: The authors have successfully answered to all my comments. Therefore, I recommend to accept this study for publication.

7. PLOS authors have the option to publish the peer review history of their article (what does this mean? ). If published, this will include your full peer review and any attached files.

**Do you want your identity to be public for this peer review?** For information about this choice, including consent withdrawal, please see our Privacy Policy .

Reviewer #2: **Yes: ** Juuso O. Hakala

---

## [Editor Report · Acceptance letter]

PONE-D-25-11838R1

PLOS ONE

Dear Dr. Varjonen,

I'm pleased to inform you that your manuscript has been deemed suitable for publication in PLOS ONE. Congratulations! Your manuscript is now being handed over to our production team.

Kind regards,

on behalf of

Dr. Gareth Hagger-Johnson

Academic Editor

PLOS ONE